# Improving recognition of common mental health disorders in Cambodia: Validation of the PHQ-9 and GAD-7 and development of a brief mental health screener

**Lesley Steinman**[1]*, **Oudamsambath Phal**[2], **Ramy Srou**[2], **Kimkanika Ung**[2], **James LoGerfo**[3,4], **Richard C. Veith**[4,5], **Tracy W. Harachi**[6]

**1** Department of Health Systems and Population Health, University of Washington School of Public Health, Seattle, Washington, United States of America, **2** Department of Social Work, Royal University of Phnom Penh, Phnom Penh, Cambodia, **3** Department of Medicine, University of Washington Schools of Medicine, Seattle, Washington, United States of America, **4** Department of Global Health, University of Washington Schools of Medicine and Public Health, Seattle, Washington, United States of America, **5** Department of Psychiatry and Behavioral Sciences, University of Washington School of Medicine, Seattle, Washington, United States of America, **6** School of Social Work, University of Washington, Seattle, Washington, United States of America

☯ These authors contributed equally to this work.
* lesles@uw.edu

## Abstract

Low- and middle-income countries (LMICs) bear 80% of the burden from common mental health disorders like depression and anxiety. One LMIC, Cambodia, has a mental health care gap due to history of genocide and civil war, persistent poverty, and under-resourced mental health system. Collaborative care models can improve access to and quality of mental health care by integrating care into existing health care infrastructure. While tools like the PHQ-9 and GAD-7 are widely used to detect unmet mental health needs and provide measurement-based care, their validity has not been established for many LMICs. We therefore sought to validate the PHQ-9 and GAD-7 and develop a brief mental health screener in the Cambodian context. This study was guided by an advisory committee of local mental health, health care, social work, and public health policymakers, practitioners, and researchers. The PHQ-9, GAD-7, and C-SSI were used to identify possible screening items and the HSCL and HTQ were used as criterion instruments to detect depression, anxiety and trauma. Pearson's correlation was used to assess validity and k-fold cross validation was used to identify which combination of items were most predictive of mental health disorders. Among 498 patients (mean(SD) age 51(12.8) years, 48.6% women, 74.4% married), 19% and 13% screened positive for depression and for anxiety using the PHQ-9 and the GAD-7, respectively. Both the PHQ-9 and GAD-7 were strongly correlated with the HSCL and HTQ. The combination of two PHQ-9 items (sleep, worthlessness) and one GAD-7 item (nervousness) most accurately predicted mental health conditions (AUC = 0.980). The PHQ-9 and GAD-7 were valid in Cambodia, and we identified several items to screen for emotional distress. We are currently using these tools in a collaborative care model pilot in urban and rural settings and welcome others application to support Cambodian mental health equity.

**Data availability statement:** This study was reviewed and approved by the Cambodian National Ethics Committee for Health Research (#068 NECHR)) and the University of Washington (UW) Institutional Review Board (#00006933). We did not request to share these data when these applications were submitted initially in 2018 as this is not appropriate in Cambodia research protocols. We no longer have participant contact information to be able to ask for permission to share data at this time. Any queries regarding data access can be directed to these non-author point of contacts at these ethics boards: NECHR at (855-23) 880-345 or the UW IRB Committee J at hsdteamj@uw.edu or (1) 206-543-0098.

**Funding:** This research was supported by the University of Washington Global Innovation Fund (GIF) (TH, LS) https://www.washington.edu/globalaffairs/gif/, the Center for Southeast Asia and Its Diasporas (CESAD) (JL, LS) https://jsis.washington.edu/csead/, and the Louvain Coopération au Développement (OP, RS, UK) https://louvaincooperation.org/en who received funding from the Else Kröner Fresenius Stiftung (EKFS) https://www.ekfs.de/ and the Directorate-General for Development Cooperation (DGD) https://diplomatie.belgium.be/en/about-us/directorate-general-development-cooperation-and-humanitarian-aid-dgd. OP, RS and UK received support from the UW School of Social Work (SSW) https://social-work.uw.edu/ and from the Royal University of Phnom Penh Department of Social Work https://rupp.edu.kh/fssh/social_work/. The funders had no role in study design, data collection and analysis, decision to publish, or preparation of the manuscript.

**Competing interests:** The authors have declared that no competing interests exist.

## Introduction

The global mental health burden is increasingly recognized as a major public health and human rights issue. Mental health was recently recognized as a "global public good" key to sustainable development in all countries [1]. Common mental health disorders like depression and anxiety impact one in four people around the world, resulting in lost health, misery, early death, and economic burden [2]. A leading cause of disability worldwide [3], 80% of the mental illness burden falls on people in low- and middle-income countries (LMIC) [4]. However, four in five people living with mental disorders in LMICs receive no treatment for their mental health conditions [5]. It is imperative that we find better ways to improve mental health in LMICs as a health and human right.

Cambodia is an LMIC under-represented in global mental health research. For instance, recent reviews of brief screening tools [6] and of interventions [7] found no studies in Cambodia. Cambodia remains deeply impacted by the multiple legacies of genocide, civil war, and colonization. Between 1969-73, the US government dropped over 110,000 bombs on Cambodia which destabilized the country. The Khmer Rouge revolution (the "Pol Pot years") in the 1970s resulted in the genocide of a quarter of the population (~ 2 million people or even more) through starvation, disease, forced labor, exhaustion, and mass extermination [8], and the struggle continued into the 1980s during a decade of war with Vietnam [9,10]. Presently, one in four women suffer from probable anxiety disorders [11] and one in six suffer from probable depressive disorders [12]. Probable estimates of PTSD rates range from 14.2% to 33.4% among Khmer Rouge survivors, while worldwide prevalence of PTSD is estimated at less than 0.4% [13]. The burden of mental health needs are great given Cambodia's historical context.

There is an enormous need to increase services for mental health in Cambodia. Like many LMICs, Cambodia extends minimal resources towards mental health services [13]. It is estimated only 0.02% of the entire Cambodian health budget goes to mental health [14]. Deprived of resources, Cambodia's mental health services remain dwarfed by the scope of the population's unmet mental health needs. Estimates suggest there are approximately 60 psychiatrists in Cambodia to serve 16 million residents [15], placing Cambodia at one of the lowest psychiatrist:patient ratios in the world [16]. According to experts who identified the Grand Challenges for Global Mental Health [1] and a recent situational analysis in Cambodia, a critical first step to improving access to care is task-shifting to integrate mental health services into routine primary health care and developing an efficient triage system based on severity of condition.

A collaborative care model seeks to engage patients who are already utilizing primary care services to identify those in need of mental health services. In the U.S. and around the world, the two-item Patient Health Questionnaire (PHQ-2) [17] and two-item Generalized Anxiety Disorder (GAD-2) [18] are often used for screening purposes, and the nine-item PHQ (PHQ-9) [19] and seven-item GAD (GAD-7) [20] are used for assessment and monitoring. Primary screening is an essential first step to reducing mental health burden as it can improve recognition of people living with mental health conditions that can be alleviated through care. Since the symptoms and stigma from mental health conditions hinder treatment seeking, primary screening in settings like health centers provide one pathway to increase recognition.

However, the PHQ-9 and GAD-7 have not been validated in Cambodia where mental health features can vary from Western constructs [21] A recent review of validation studies in LMICs found 66 studies from Asia, only 18 of which were from Southeast Asia (Malaysia, Thailand, and Vietnam) with six other Southeast Asian countries including Cambodia not represented [6]. Critical is the need to use measurement tools which have demonstrated

cross-cultural equivalence as these tools guide both identification of mental distress and disorders and are employed to monitor and evaluate treatment outcomes [22–24].

Hence, this study sought to a) validate the PHQ-9 and GAD-7 within the Cambodian context; and b) identify items to develop a short screening tool for emotional distress.

## Methods

### Setting

Cambodia is an LMIC in Southeast Asia. As of 2021 the Cambodian population is 16.6 million people [25] and is largely homogenous with over 95% identifying ethnically as Khmer or Cambodian [3]. Cambodia reached LMIC status in 2015 according to World Bank classifications with a per capita gross national income GINI of $1,070 [26]. This change in designation from low to lower-middle income status belies the fact that 40% of the population continues to live on $2 or less per day, and 80% of Cambodians live in rural areas with limited access to adequate living conditions, economic opportunities and health care [3]. Educational attainment is on average five years, with most employed persons working in agricultural, industry and service sectors, and almost half (48%) working in informal jobs [26].

### Consulting committee

This study was led by the Department of Social Work (DSW) at the Royal University of Phnom Penh with guidance from an advisory committee of U.S. and Cambodian mental health experts. In June 2018, a small group of multidisciplinary U.S. and Cambodian experts were convened to participate in a workshop convened by the Ministry of Health's Department of Mental Health and Substance Abuse (MoH/DMHSA) and co-sponsored by the University of Washington/Royal University of Phnom Penh (UW RUPP) Partnering For Health initiative and the MoH/DSMHSA. Participants included physicians, psychiatrists, social workers, faculty from departments of psychology and social work, and staff from MoH/DMHSA. All agreed that general practice physicians have limited time and it would be useful to institute a short screening tool to facilitate the recognition and treatment of mental health concerns. This committee was reconvened in May 2023 (delayed due to the COVID-9 pandemic) to review findings from this study and make recommendations for practice.

### Ethics approval and inclusivity in global research

An ethics application was submitted to the Cambodia Ministry of Health Ethics Board as well as to the University of Washington and approval was granted by both institutional review systems. In addition, the Cambodia-China Friendship Preah Kossamak Hospital approved this study to take place at their center. Additional information regarding the ethical, cultural, and scientific considerations specific to inclusivity in global research is included in the Supporting Information (S1 Checklist).

### Translation

We used several recommended best practices for instrument translation [27]. All study instruments were translated into Khmer by the bilingual study directors at RUPP DSW beginning "with tools used by the Transcultural Psychological Organization (TPO). TPO is an international NGO established to provide mental health care in 1995 and in 2000 registered as an independent NGO run and staffed by Cambodians. The DSW team then reviewed and back translated items with mental health professionals in Cambodia and in the U.S., and edited several items to better align with intended meaning and to improve cultural appropriateness.

## Measures

**PHQ-9 and GAD-7 validation.** For the study, the committee decided to validate the PHQ-9 [19] and GAD-7 [20] given their widespread use as assessment tools for depression and anxiety. The total score on the PHQ-9 can be used to diagnose and assess the level of clinical depression severity: no reported symptoms (0–4), mild symptoms (5-9), moderate (10-14), moderately severe (15-19), and severe (20-24). The committee chose to validate the PHQ-9 over the PHQ-8 given the importance of asking the 9th item on suicidal ideation as a first step for suicide prevention. The GAD-7 uses these categories for symptom severity: no reported symptoms (0–4), mild symptoms (5-9), moderate (10-14), severe (15-21). The accuracy of the PHQ-9 and GAD-7 were tested against two criterion instruments, the 25-item Hopkins Symptom Checklist (HSCL-25) [28] for depression and anxiety disorders and the 16-item Harvard Trauma Questionnaire (HTQ) [29] for trauma/PTSD. The HTQ was selected in addition to the HSCL given the overlap between depression, anxiety and trauma/PTSD in the Cambodian context and that the HSCL was developed for a Western context. As such we were advised by our Cambodian research and practice colleagues to use both the HSCL and HTQ measures as criterion. Both of the HSCL-25 and the HTQ have been found to be reliable and validated with the target population [29–31]. Within our study sample, the alpha reliability for the two criterion instruments were 0.94 for the HTQ and 0.93 for the H-SCL, which are similar to those reported in other studies [30]. The recommended threshold for the HTQ is a score of trauma symptoms greater than or equal to 2.0 [12,29,32]. For the HSCL-25, the recommended threshold is greater than or equal to 1.75 [33].

**Brief emotional health screener.** The World Health Organization (WHO) defines mental health as the ability to realize one's potential, cope with normal stress, work productively and fruitfully, and to contribute to one's community [34]. Poor mental health can be defined by both cultural idioms of distress (CIDs) that describe the experience of illness distinct across cultures [35], and by common mental disorders (CMDs) which are emotional or behavioral disorders that cause distress and functional impairment [36]. As such, for the screeners the committee decided to examine items from both CID and CMD measures. The Cambodian Somatic and Syndrome Inventory (C-SSI) [37] was developed to identify key cultural idioms of distress in Cambodia [38–40]. The consulting committee agreed that the study should examine three C-SSI items (neck soreness, dizziness, and thinking too much) as common indicators of emotional distress. All items from the PHQ-9 and GAD-7 were included to capture CMDs. To develop the screener, the possible screening items were tested against two criterion instruments, the 25-item Hopkins Symptom Checklist (HSCL-25) [28] for depression and anxiety disorders and the 16-item Harvard Trauma Questionnaire (HTQ) [29] for trauma/PTSD.

Table 1 lists items from the criterion instruments HSCL-25 and HTQ and Table 2 provides a list of potential screening items from the PHQ-9, GAD-7, and C-SSI. These items were all included in the participant interviews. All items in English and in Khmer are provided in S1 Appendix.

## Sample

Given the interest to develop a tool useful within health care settings, patients coming to the Cambodia-China Friendship Preah Kossamak Hospital's Cambodia-Korea Diabetes Center and Mental Health Unit were recruited as participants for the study. Patients in both the Center and the Unit come before opening hours and wait in the lobby on a first-come, first-served process. Trained interviewers sought out patients in the lobby and invited them to participate. Participants were given a nominal cash ($2) thank you for consenting and proceeding with

**Table 1. List of criterion instrument items from the H-SCL D/A and HTQ.**

| Source | Item |
| --- | --- |
| | *Please carefully decide how much these things bothered you in the PAST WEEK……* |
| **H-SCL D** | |
| | Feeling low in energy, slowed down. |
| | Blaming yourself for things. |
| | Crying easily. |
| | Loss of sexual interest or pleasure. |
| | Poor appetite. |
| | Difficulty falling asleep, staying asleep. |
| | Feeling hopeless about the future. |
| | Feeling sad. |
| | Feeling lonely. |
| | Thoughts of ending your life. |
| | Feeling of being trapped or caught. |
| | Worrying too much about things. |
| | Feeling no interest in things. |
| | Feeling everything is an effort. |
| | Feelings of worthlessness. |
| **H-SCL A** | |
| | Faintness, dizziness, or weakness. |
| | Feeling fearful. |
| | Feeling restless, can't sit still. |
| | Feeling tense or keyed up. |
| | Headaches. |
| | Heart pounding or racing. |
| | Nervousness or shakiness inside. |
| | Spells of terror or panic. |
| | Suddenly scared for no reason. |
| | Trembling. |
| **HTQ-Section IV** | |
| | Recurring thoughts or memories of the most hurtful or terrifying events. |
| | Feeling that you have no one to rely on. |
| | Feeling as though the hurtful or terrifying event is happening again. |
| | Finding out or being told by other people that you have done something that you cannot remember. |
| | Recurrent nightmares. |
| | Feeling as if you are split into two people and one of you is watching what the other is doing. |
| | Feeling detached or withdrawn from people. |
| | Feeling someone you trusted betrayed you. |
| | Unable to feel emotions. |
| | Feeling jumpy or easily startled. |
| | Difficulty concentrating. |
| | Trouble sleeping. |
| | Feeling on guard. |
| | Feeling irritable or having outburst of anger. |
| | Avoiding activities that remind you of the traumatic or hurtful event. |
| | Inability to remember parts of the most traumatic or hurtful events. |
| | Less interest in daily activities. |
| | Feeling as if you don't have a future. |

*(Continued)*

**Table 1.** (Continued)

| Source | Item |
|---|---|
| | Avoiding thoughts or feelings associated with the traumatic or hurtful events. |
| | Sudden emotional or physical reaction when reminded of the most hurtful or traumatic events. |
| | Feeling that people do not understand what happened to you. |
| | Difficulty performing work or daily tasks. |
| | Blaming yourself for things that have happened. |
| | Feeling guilty for having survived. |
| | Feeling hopelessness. |
| | Feeling ashamed of the hurtful or traumatic events that have happened to you. |
| | Spending time thinking about why these things happened to you. |
| | Feeling as if you are going crazy. |
| | Feeling that you are the only one who suffered these events. |
| | Feeling others are hostile toward you. |

HTQ = Harvard Trauma Questionnaire, H-SCL = Hopkins Symptom Checklist, D = Depression, A = Anxiety

**Table 2. Potential screening items from the PHQ-9, GAD-7, C-SSI and descriptive statistics.**

| Source | Item | Min | Max | Mean | SD | N |
|---|---|---|---|---|---|---|
| **PHQ-9** | | | | | | |
| | 1. Little interest or pleasure in doing things* | 0 | 3 | 0.64 | 0.83 | 495 |
| | 2. Feeling down, depressed, or hopeless* | 0 | 3 | 0.70 | 0.91 | 497 |
| | 3. Trouble falling, staying asleep, or sleeping too much | 0 | 3 | 1.01 | 1.05 | 498 |
| | 4. Feeling tired or having little energy | 0 | 3 | 1.12 | 0.82 | 497 |
| | 5. Poor appetite or overeating | 0 | 3 | 0.78 | 0.91 | 498 |
| | 6. Feeling bad about oneself or that you are a failure or make yourself or down your family | 0 | 3 | 0.47 | 0.79 | 497 |
| | 7. Trouble concentrating on things, such as reading the newspaper or watching television | 0 | 3 | 0.53 | 0.80 | 497 |
| | 8. Moving or speaking so slowly that other people could have noticed? Or the opposite being so fidgety or restless that you have been moving around a lot more than usual.. | 0 | 3 | 0.46 | 0.80 | 495 |
| | 9. Thoughts that you would be better off dead or of hurting yourself in someways…** | 0 | 3 | 0.16 | 0.50 | 497 |
| **GAD-7** | | | | | | |
| | 1. Feeling nervous or on edge | 0 | 3 | 0.54 | 0.76 | 496 |
| | 2. Not being able to stop or control worrying | 0 | 3 | 0.65 | 0.82 | 496 |
| | 3. Worrying too much about different things | 0 | 3 | 1.06 | 0.99 | 496 |
| | 4. Trouble relaxing | 0 | 3 | 0.62 | 0.79 | 498 |
| | 5. Being so restless that it is hard to sit still** | 0 | 3 | 0.57 | 0.79 | 498 |
| | 6. Becoming easily annoyed or irritable | 0 | 3 | 1.25 | 0.94 | 497 |
| | 7. Feeling afraid as if something awful might happen | 0 | 3 | 0.40 | 0.75 | 498 |
| **C-SSI** | | | | | | |
| | Thinking lots | 0 | 3 | 1.26 | 1.10 | 497 |
| | Dizziness | 0 | 3 | 0.60 | 0.85 | 497 |
| | Neck soreness | 0 | 3 | 0.60 | 0.86 | 497 |

C-SSI = Cambodia Symptom and Syndrome Inventory, GAD-7 = Generalized Anxiety Disorder 7-item, PHQ-9 = Patient Health Questionnaire 9-item, SD = Standard Deviation, N = number of participants, Min = minimum value, Max = Maximum value. *These two items make up the PHQ-2 screener. **These two items were not included in the possible screening item pool due to concerns about invasiveness and multicollinearity.

the interview. Patients were eligible if they were 18 years old or above, able to speak Khmer, and able to complete the informed consent process. Participants with mental health distress including suicidal ideation were referred for mental health services.

## Data collection

One-on-one participant interviews were conducted with data entered into an electronic form. The DSW recruited social work students and recent graduates to be research assistants for data collection. Training of research assistants covered gaining rapport with participants, gathering informed consent, using appropriate strategies to collect quality data via an in-person interview, and entering responses into a Wi-Fi-connected tablet. If a patient endorsed the suicidal ideation item of the PHQ-9, they were referred to the on-site social worker. While interviewers were trained on procedures to alert the in-country research coordinator if the patient became distressed and to refer participants to the on-site social worker, this situation did not occur. There were no negative incidents to report. The consent rate was 89.9%, affected by a minor number of refusals; a few consenting patients were not able to complete the interview given their available time and were not included hence the final sample is N = 498.

## Data analysis

We used SPSS version 26 for data analysis [41]. Descriptive statistics were used to report the prevalence of each item and total scale scores. Validation of the PHQ-9 and GAD-7 was conducted using pairwise correlations between total scores of the PHQ-9 and GAD-7 with the two criterion instruments (HSCL-25 and HTQ).

We used four steps to create and evaluate a brief screener of emotional distress – 1) identify a sample of distressed patients, 2) identify possible screening items, 3) test the accuracy of these screening items, and 4) identify appropriate thresholds for the screener. We first identified who in the sample met criteria for trauma, depression or anxiety using the HTQ>/= 2 or HSCL-25>/= 1.75. Since the study was interested in a general emotional or psychological distress screener rather than symptoms of a specific disorder, e.g., depression, respondents who met either threshold for the HTQ (>/= 2.0) or HSCL-25 (>/= 1.75) were recoded into a new combined emotional distress variable.

Second, given our large potential pool of items, we used a forward stepwise logistic regression to identify possible screening items that were significantly related to being emotionally distressed. The combined emotional distress variable was entered as an outcome and a set of 17 screening items were included as possible screeners. Note that the original list of potential items included 19 total items, however we excluded 2 items from the variable selection process: the 9th PHQ-9 item, "Thoughts that you would be better off dead or of hurting yourself in some ways…," was not included as the goal was to identify a broad screen for distress, and asking about suicidal thoughts within an initial assessment was considered too invasive. The GAD-7 item, "Being so restless that it is hard to sit still," was also excluded to reduce issues related to multicollinearity with the PHQ-9 8th item "…being so fidgety or restless that you have been moving around a lot more than usual."

The third step was to test the accuracy of the different combinations of screening items to predict whether a person had clinically significant trauma, depression or anxiety. An important feature of predictive modelling is the ability of a model to generalize to new cases. Evaluating the predictive performance (Area Under the Curve, or AUC) of a set of screening items using all cases from the original analysis sample can yield too optimistic an estimate of predictive performance. We used K-fold cross-validation to generate a more realistic estimate of predictive performance. This method for cross-validation is also

recommended when the number of observations is not very large. After fitting the binary logistic regression model with a set of independent variables, the predictive performance of this set of variables was assessed by the area under the curve (AUC). We estimated the AUC for a sample (the 'test' sample) that was independent of the sample used to predict the dependent variable (the 'training' sample). AUCs range from 0 to 1, with higher AUCs indicating that each set of screening items is better able to distinguish between patients with mental health disorders and those without.

The fourth step identified appropriate thresholds or cut offs for the proposed screeners. Receiver operating characteristic (ROC) analysis was used for comparing predictive models and is often used in clinical medicine and social science to assess the trade-off between model sensitivity (probability that the screener will indicate distress with positive case) and specificity (probability of the screener to correctly generate a negative result for those who don't have distress) results.

## Results

### Demographics

The 498 participants in the study ranged in age from 17 to 86 years (mean(SD) 51(12.8) years) and 51.4% (N = 256) were male. The majority of respondents reported being married (74.4%); 13.1% were widowed, 10.3% single, and 2.2% separated or divorced. Eighty-two percent (82.5%) were from the diabetes clinic and 17.5% were from the mental health unit. When asked whether they had ever sought mental health services in the past, 18.9% reported affirmatively. Women were significantly more likely to have reported seeking services (64 out of 242 or 26.4%) in contrast to men (11.7%).

### Descriptive information on potential screening items

Table 2 reports descriptive statistics for each of the potential screening items. The PHQ-9 items with the highest means were "trouble falling, staying asleep or sleeping too much" and "feeling tired or having little energy," while the GAD-7 items with highest means were "becoming easily annoyed or irritable" and "worrying too much about different things." "Thinking lots" was highest among the C-SSI items. Of note, 11.2% of the sample reported some suicidal thoughts in the PHQ-9 item.

### Validating the PHQ-9 and GAD-7 within the Cambodian context

**Descriptive PHQ-2, PHQ-9 and GAD-7 data.** Primary care providers typically use the first two items in the PHQ-9 for screening (called the PHQ-2). Scores of 3 or greater warrant implementation of the other seven items to capture a total depression score. Within our study sample, 100 patients or 20.2% would have warranted further assessment based on their PHQ-2 score. Looking at the full study sample, the distribution of patients by standard cut offs for depression severity was 19% reporting clinically significant depression (PHQ-9>/10 cut-off) comprised of 13.1% moderate symptoms, 4.9% moderately severe symptoms, and 1% severe symptom levels; 36.3% had mild symptoms.

When we compared the subset who had a score of 3+ on the PHQ-2 and would have warranted further assessment with these categories, we noted that the PHQ-2 as a screening tool with this sample would have missed 8.4% of patients who might benefit with further assessment if the PHQ2 was relied upon. Looking at the GAD-7, the distribution of symptoms found that 13% had clinically significant anxiety: 8.7% had moderate anxiety and 4.3% had severe anxiety.

### Relationship of PHQ-9 and GAD-7 to criterion instruments

The relationship between the total PHQ-9 score and GAD-7 total score in our study was quite strong (r = 0.798, p <.001). In terms of how well the PHQ-9, and GAD-7 total scores were associated with the two criterion instruments, the correlations were all highly significant (p <.001). With the HSCL-25, the correlations were 0.799 with the PHQ-9, and 0.831 with the GAD-7. With the HTQ, the correlations were 0.544 with the PHQ-2, 0.766 with the PHQ-9, and 0.813 with the GAD-7. Table 3 displays the correlation matrix.

### Developing a short screening tool for emotional distress

**Identifying screening candidates.** Nine (1.8%) and 21 (4.2%) of respondents met the threshold for trauma (HTQ>/=2) or depression or anxiety (HSCL-25>/=1.75) with all respondents who had clinically significant trauma symptoms endorsing clinically significant depression and anxiety symptoms as well. As such, 21 (4.2%) of the respondents were coded "1" to indicate clinically significant emotional distress, and the remainder were coded "0" to indicate no emotional distress. Findings from the forward stepwise logistic regression to identify possible screening items are provided in Tables 4 and 5. These tables provide the model summary tests for each step within the regression and highlight which variables were entered with each of the steps. These six sets of screening items were then used in our subsequent k-fold cross validation analyses to evaluate the ability of each set to predict trauma/PTSD, depression or anxiety using the AUC as our measure of predictive performance.

### Testing the accuracy of the screening candidates

Based on our k-fold cross-validation, Step 7 (five screening items) had the highest AUC of 0.988. The next best set of screening items is Step 4 which has a similar AUC of 0.985. Four items are more parsimonious for a brief screening instrument for emotional distress and yields a predictive accuracy of mental health disorder that is very close to the accuracy yielded by asking seven screening items. Other possible sets of screening items include Step 5 – in which 3 of the 4 items from Step 4 had an AUC of 0.980 - and Step 6, which had an AUC of

**Table 3. Correlations between the PHQ-9 and GAD-7 and the criterion instruments (HTQ and HSCL-25).**

| Source | Correlations, Significance, N | HTQ | H-SCL | GAD-7 | PHQ-9 |
|---|---|---|---|---|---|
| **HTQ** | Pearson Correlation | 1 | .889 | .813 | .760 |
| | Significance (2-tailed) | | .000 | .000 | .000 |
| | N | | 455 | 451 | 447 |
| **H-SCL** | Pearson Correlation | | 1 | .831 | .799 |
| | Significance (2-tailed) | | | .000 | .000 |
| | N | | | 485 | 481 |
| **GAD-7** | Pearson Correlation | | | 1 | .798 |
| | Significance (2-tailed) | | | | .000 |
| | N | | | | 485 |
| **PHQ-9** | Pearson Correlation | | | | 1 |
| | Significance (2-tailed) | | 1 | .831 | .799 |
| | N | | | .000 | .000 |

GAD-7 = Generalized Anxiety Disorder 7-item, HTQ = Harvard Trauma Questionnaire, H-SCL = Hopkins Symptom Checklist, PHQ-9 = Patient Health Questionnaire 9-item, N = number of participants.

**Table 4. Model summary and items identified in the forward stepwise logistic regression.**

| Step | -2 Log likelihood | Cox & Snell R Square | Nagelkerke R Square |
|---|---|---|---|
| 1 | 97.809[a] | .132 | .455 |
| 2 | 75.248[b] | .172 | .591 |
| 3 | 62.829[c] | .193 | .663 |
| 4 | 54.799[c] | .206 | .708 |
| 5 | 56.649[c] | .203 | .698 |
| 6 | 51.360[c] | .212 | .728 |
| 7 | 46.455[d] | .220 | .755 |

[a]Estimation terminated at iteration number 7 because parameter estimates changed by less than.001.

[b]Estimation terminated at iteration number 8 because parameter estimates changed by less than.001.

[c]Estimation terminated at iteration number 9 because parameter estimates changed by less than.001.

[d]Estimation terminated at iteration number 10 because parameter estimates changed by less than.001.

**Table 5. Items identified in the logistic regression and accuracy of each set of screening items (AUCs).**

| Source | Item | Step 1 | Step 2 | Step 3 | Step 4 | Step 5 | Step 6 | Step 7 |
|---|---|---|---|---|---|---|---|---|
| PHQ-2 | Little interest or pleasure in doing things | | | | | | | |
| PHQ-2 | **Feeling down, depressed, or hopeless** | | | | | | 6 | 7 |
| PHQ-9 | **Trouble falling, staying asleep, or sleeping too much** | | | | | | | 7 |
| PHQ-9 | **Feeling tired or having little energy** | | | | 4 | 5 | 6 | 7 |
| PHQ-9 | Poor appetite or overeating | | | | | | | |
| PHQ-9 | **Feeling bad about oneself or that you are a failure or make yourself or down your family** | | | 3 | 4 | 5 | 6 | 7 |
| PHQ-9 | Trouble concentrating on things, such as reading the newspaper or watching television | | | | | | | |
| PHQ-9 | **Moving or speaking so slowly that other people could have noticed? Or the opposite being so fidgety or restless that you have been moving around a lot more than usual..** | | 2 | 3 | 4 | 5 | 6 | 7 |
| PHQ-9* | Thoughts that you would be better off dead or of hurting yourself in someways… | | | | | | | |
| GAD-7 | **Feeling nervous or on edge** | 1 | 2 | 3 | 4 | | | |
| GAD-7 | Not being able to stop or control worrying | | | | | | | |
| GAD-7 | Worrying too much about different things | | | | | | | |
| GAD-7 | Trouble relaxing | | | | | | | |
| GAD-7* | Being so restless that it is hard to sit still | | | | | | | |
| GAD-7 | Becoming easily annoyed or irritable | | | | | | | |
| GAD-7 | Feeling afraid as if something awful might happen | | | | | | | |
| C-SSI | Thinking lots | | | | | | | |
| C-SSI | Dizziness | | | | | | | |
| C-SSI | Neck soreness | | | | | | | |
| | **AUCs** | 0.962 | 0.968 | 0.972 | 0.985 | 0.980 | 0.982 | 0.988 |

AUC = Area Under the Curve, C-SSI = Cambodia Symptom and Syndrome Inventory, GAD-7 = Generalized Anxiety Disorder 7-item, PHQ-9 = Patient Health Questionnaire 9-item, N = number of participants, * Not included

0.982 and includes the second item of the PHQ-2 (depressed mood) but removes the first item of the GAD-7 (being nervous or on edge). The later item was the leading item candidate in Step 1 and present in 3 other steps. We also ran the k-fold cross validation for the PHQ-2 as this is a commonly used screening measure for depression. The AUC was 0.916, suggesting

that the combination of screening items identified and tested in this study are stronger screeners of emotional distress than relying on the PHQ-2 within our study sample.

### Identifying thresholds or cut points for proposed screening items

It is recommended that a score of 3+ on the PHQ2 is the threshold or cut point suggestive of clinically significant depression. Looking at the three possible screeners in this study, the distribution of total scores for Step 4 (4 items) is 0-12 ($\bar{x}$ = 2.58, SD = 2.25), for Step 5 (3 items) is 0-9 ($\bar{x}$ = 2.04, SD = 1.74), and Step 6 (4 items) is 0-12 ($\bar{x}$ = 2.74, SD = 2.33). ROC analyses with the full sample provided the following sensitivity (probability that the screener will indicate distress with positive case) and specificity (probability of the screener to correctly generate a negative result for those who don't have distress) results.

| Possible Screeners | Score | Sensitivity | 1-Specificity | Suggested Cut Point |
|---|---|---|---|---|
| Step 4 – 4 items | 5.5 | 1.0 | 0.07 | 6+ |
| Step 5 – 3 items | 3.5 | 1.0 | 0.14 | 4+ |
| Step 6 – 4 items | 6.5 | 1.0 | 0.04 | 7+ |

## Discussion

We found that the PHQ-9 and GAD-7 were valid in the Cambodian context and can continue to be used as instruments to assess for depression and anxiety. The PHQ-2 did not perform as well as other sets of items as a screener for broad emotional distress. Several items were identified and accurate as a brief screener of emotional distress. Combinations of items identified in Step 4, 5, or 6 would make appropriate brief screening tools with a threshold score of 6 or greater warranting additional assessment using the remaining PHQ-9 and GAD-7 items.

The Advisory Committee reviewed the study findings to make recommendations for practice in Cambodia. The committee included researchers and practitioners from diverse settings; as such their recommendations were helpful to translating study findings into more generalizable and feasible practice guidance. First, while step 7/the seven-item screener had the highest AUC, this set of items was not selected as adding seven questions to patient screening was deemed not feasible in busy health care clinics where there is both limited staffing and time. Furthermore, concern was expressed about item 8 on the PHQ 9 (Moving or speaking so slowly that other people could have noticed? Or the opposite being so fidgety or restless that you have been moving around a lot more than usual) because of its complexity. In addition, Cambodian clinicians expressed that patients often present with a chief complaint about sleep issues which while related to the low energy item capture a specific somatic symptom of emotional distress in Cambodia. We therefore re-ran the analyses without complex item 8, and substituted item 3 about sleep for item 4 about low energy. The results showed no significant loss of predictive values or AUCs suggesting similar accuracy and more feasibility for future practice. Therefore, we chose to use the resulting 3-item screener (S2 Appendix) for piloting in our current collaborative care research with patient care facilitators at the National Cancer Center in urban Phnom Penh and in Chamkar Leu Referral Hospital in rural Kampong Cham province.

Screening is an important population health strategy for improving the detection of common mental health disorders in LMICs by identifying people who may have CMDs but they or their health care provider are not aware of or seeking or providing care for them [42]. Our study adds to the growing literature by building the evidence base for mental health screening and assessment in Cambodia. It is not surprising that our screener included somatic complaints as these are the most common worldwide for common mental disorders

[43]. While we included cultural idioms of distress given their potential to be less stigmatizing and more culturally acceptable and accurate given high false positives [44], none of these ultimately were predictive of depression, anxiety or trauma. This may be because though used widely, our criterion instruments may not have fully captured cultural idioms of distress and as such are limited in identifying clinically significant mental health disorders in the Cambodian context.

As mentioned in the introduction, better screening and assessment is key for closing the mental health care gap and improving mental health – in particular, adapting screening for local contexts and incorporating into health care in LMICs to help reduce societal burden and costs from disease and disability [45]. However, since screening without treatment and follow-up is unethical [46], we are currently building upon our partnership for this study to adapt and deliver a collaborative care model (CCM) to improve access to quality mental health care in two rural Cambodian provinces. CCM was developed at the UW and aligns with global mental health field's urgent call to address inequities and close the treatment gap in resource-constrained settings by building capacity among non-mental health providers to provide effective team-based care [1,47,48]. CCM can improve access to quality mental health care by expanding availability of care in settings with limited access, workforce gaps, and stigma [49,50]. Delivered by care managers, CCM can also address unmet social needs that drive health outcomes like poor housing and food insecurity [51]. Since 80% of Cambodians live in rural areas with limited access to adequate living conditions, economic opportunities and health care [3]; adapting CCM for this context offers tremendous opportunity to improve health equity. While initially developed for highly resourced clinical settings [52,53], CCM evidence is emerging in resource-constrained settings in Asia including Nepal, Vietnam, and India [54–56].

Our study aligns with several recent recommendations for developing and evaluating mental health screening and assessment measures in resource-constrained settings [57]. We attended to the Cambodian context by hiring and training data collectors from the country of origin and including local practitioners, policymakers and researchers in reviewing, selecting, translating and adapting measures. We convened an advisory group to qualitatively evaluate the adapted measures and quantitatively assessed measure performance. We administered the measures verbally given low education and literacy in Cambodia and cultural preferences, and evaluated the psychometrics of these measures that were initially designed to be written measures. Lastly, we provided information about how the measures were translated and adapted for the Cambodian population.

There are also several limitations and opportunities for future research. First, while Khmer is the dominant language in Cambodia, we did not assess for language fluency. As such, the measures might underserve people who do not speak Khmer and who speak other preferred dialects such as Cham. We did not collect inter-rater reliability data due to time and resource constraints. Future research should examine inter-rater reliability given that the measures are being administered verbally and small variations in how the items are phrased or in patient's body language may influence scoring, thus presenting an opportunity to improve training for enhanced instrument fidelity. Furthermore, while we recruited patients from both the Diabetes Center and Mental Health Unit to include a range of mental health symptoms, we might have introduced information bias to our study results given the heterogeneity in data collection settings.

In closing, our study brought together key Cambodian stakeholders from the research, policy and practice sectors to validate existing mental health screening and assessment instruments and to identify and test a brief screener of emotional distress. Study findings suggest these tools are acceptable and feasible in the Cambodian context. This research is essential for

closing the mental health treatment gap by improving the recognition of clinically significant mental health disorders in primary care and care for non-communicable diseases.

## Supporting information

**S1 Appendix. Study measures.**
(DOCX)

**S2 Appendix. Final 3-item screener for emotional distress in Cambodia.**
(DOCX)

**S1 Checklist. Inclusivity in global research.**
(DOCX)

## Acknowledgments

We thank Dr. Chhit Sophal, Director of the Department of Mental Health and Substance Abuse (DMHSA) for his encouragement and support of the need for a screening tool, for convening the initial stakeholders group, and for recommending to work with the Cambodia-China Friendship Preah Kossamak Hospital. Gratitude to Dr. Touch Khun, Head of the Cambodia-Korean Diabetes Center, and Dr. Kheng Khannara, Head of the Mental Health Unit, at the Cambodia-China Friendship Preah Kossamak Hospital, for their cooperation as the study site and collaboration with the DMHSA. Thanks to the patients at the Diabetes Center and Mental Health Unit who provided survey data for this study. We wish to thank BSW students and recent graduates from the Royal University of Phnom Penh's Department of Social Work who were trained as research assistants to collect study data. Thanks to the multidisciplinary US and Cambodian experts who participated in a consultancy workshop co-sponsored by the University of Washington/Royal University of Phnom Penh (UW RUPP) Partnership/Partnering For Health and the Ministry of Health Department of Mental Health and Substance Abuse (MoH/DSMHSA) who provided invaluable information to guide study design, methods and interpretation: Kong Sokdina, Dr. Sok Hong, Dr. Chhiv Sikheang, Chou Phallyka, Dr. Sin Eap, Dr. Pen Nouth, Ouk Vannay, Heng Sovandara, Dr. Robert Brooks, Kaing Rachana, Dr. Chantha Tola, Dr. Ean Nil, Dr. Muny Sothara, Dr. Sar Sothearith, An Theary. Thank you to the Transcultural Psychology Organization (TPO) Cambodia for translating the mental health instruments into Khmer. Thank you to Dr. Lydia Chwastiak from the UW Department of Psychiatry and Behavioral Sciences for advising on study design and interpretation, and to Dr. Jennifer Velloza from UW Department of Global Health for advising on analytic methods.

## Author contributions

**Conceptualization:** Lesley Steinman, James LoGerfo, Richard C. Veith, Tracy Harachi.

**Data curation:** Lesley Steinman, Oudsambath Phal, Ramy Srou, Tracy Harachi.

**Formal analysis:** Lesley Steinman, Tracy Harachi.

**Funding acquisition:** Kimkanika Ung, James LoGerfo, Tracy Harachi.

**Methodology:** Lesley Steinman, James LoGerfo, Richard C. Veith, Tracy Harachi.

**Project administration:** Oudsambath Phal, Ramy Srou, Kimkanika Ung, Tracy Harachi.

**Resources:** Oudsambath Phal, Ramy Srou, Kimkanika Ung.

**Software:** Ramy Srou.

**Supervision:** Oudsambath Phal, Ramy Srou, Kimkanika Ung, James LoGerfo, Richard C. Veith, Tracy Harachi.

**Writing – original draft:** Lesley Steinman, Oudsambath Phal, Ramy Srou, James LoGerfo, Richard C. Veith, Tracy Harachi.

**Writing – review & editing:** Lesley Steinman, Oudsambath Phal, Ramy Srou, Kimkanika Ung, James LoGerfo, Richard C. Veith, Tracy Harachi.

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
