## [Decision Letter · Decision Letter 0]

5 Nov 2024

PMEN-D-24-00347

Improving recognition of common mental health disorders in Cambodia: Validation of the PHQ-9 and GAD-7 and development of a brief mental health screener.

PLOS Mental Health

Dear Dr. Steinman,

Thank you for submitting your manuscript to PLOS Mental Health. After careful consideration, we feel that it has merit but does not fully meet PLOS Mental Health’s publication criteria as it currently stands. Therefore, we invite you to submit a revised version of the manuscript that addresses the points raised during the review process.

There are several issues with the manuscript, as the reviewers mentioned below.  Besides the language and methodological issues, the authors **must comply** with the journal guidelines regarding data availability and statistical analysis. 

We look forward to receiving your revised manuscript.

Kind regards,

Haroon Lone

Academic Editor

PLOS Mental Health

Journal Requirements:

1. Please include a complete copy of PLOS’ questionnaire on inclusivity in global research in your revised manuscript. Our policy for research in this area aims to improve transparency in the reporting of research performed outside of researchers’ own country or community. The policy applies to researchers who have travelled to a different country to conduct research, research with Indigenous populations or their lands, and research on cultural artefacts. The questionnaire can also be requested at the journal’s discretion for any other submissions, even if these conditions are not met.  Please find more information on the policy and a link to download a blank copy of the questionnaire here: https://journals.plos.org/plosmentalhealth/s/best-practices-in-research-reporting. Please upload a completed version of your questionnaire as Supporting Information when you resubmit your manuscript.

2. In the online submission form, you indicated that "De-identified participant data is available upon request by contacting the corresponding author.". 

3. Uploaded as supplementary information.

Additional Editor Comments (if provided):

Reviewers' comments:

Reviewer's Responses to Questions

**Comments to the Author**

1. Does this manuscript meet PLOS Mental Health’s publication criteria ? Is the manuscript technically sound, and do the data support the conclusions? The manuscript must describe methodologically and ethically rigorous research with conclusions that are appropriately drawn based on the data presented.

Reviewer #1: Yes

Reviewer #2: Partly

2. Has the statistical analysis been performed appropriately and rigorously?

Reviewer #1: I don't know

Reviewer #2: Yes

3. Have the authors made all data underlying the findings in their manuscript fully available (please refer to the Data Availability Statement at the start of the manuscript PDF file)?

Reviewer #1: Yes

Reviewer #2: No

4. Is the manuscript presented in an intelligible fashion and written in standard English?

Reviewer #1: No

Reviewer #2: Yes

5. Review Comments to the Author

Reviewer #1: Thank you for including the measures-perhaps this could be added as an appendix afterwards

hurting yourself in some ways…,” was not included as the goal was to identify a broad screen

for distress, plus asking about suicidal thoughts within an initial assessment was considered too

invasive.-I would elaborate more on this and why perhaps phq-8 was not utilized instead

GAD-7- which other items for multicollinearity?

I couldn’t read table 5

This paper was very interesting-perhaps in future you can include 9th question in analysis. Please check for grammatical errors and redundancy.

Reviewer #2: This study aimed to develop a brief mental health screener applicable to the context of Cambodia, a representative low- and middle-income country (LMIC). Undoubtedly, the research holds significant public health importance, as LMICs disproportionately bear the global mental health disease burden. However, there are several major issues that need to be addressed.

1.The introduction section is lengthy regarding the mental health burden and the need for mental health services, yet it does not clearly present the importance of primary psychological screening.

2.Why did the authors choose the trauma/PSTD scale (i.e., the 16-item Harvard Trauma Questionnaire (HTQ)) as the criterion instrument to test the accuracy of the PHQ-9 and GAD-7?

3.In the methods section, the introduction of the Cambodian Somatic and Syndrome Inventory (C-SSI) appears somewhat abrupt. I recommend providing additional context to explain why the C-SSI was chosen alongside the PHQ-9 and GAD-7.

4.It would be better to provide more details on the choice of a sample size of 500 participants and the basis for estimating that a minimum of 30% will strongly endorse symptoms.

5.In the results section, why did the authors decide to omit step 7. The explanation that four items (step 4) are more feasible for a brief screening instrument for emotional distress lacks persuasiveness. Furthermore, why did the authors substitute item 3 for item 4 in the Advisory Committee Review and Recommendations section?

6.The study only recruited patients coming to the Cambodia-China Friendship Preah Kossamak Hospital’s Cambodia-Korea Diabetes Center and Mental Health Unit. This apparent heterogeneity in data collection methods may introduce information bias to their study results. This limitation should be acknowledged.

7.The authors should explain why de-identified participant data is available only upon request, given that the PLOS Data policy requires authors to make all data underlying the findings described in their manuscript fully available without restriction, except in rare cases.

6. PLOS authors have the option to publish the peer review history of their article (what does this mean? ). If published, this will include your full peer review and any attached files.

**Do you want your identity to be public for this peer review?** For information about this choice, including consent withdrawal, please see our Privacy Policy .

Reviewer #1: No

Reviewer #2: No

---

## [Decision Letter · Decision Letter 1]

20 Feb 2025

Improving recognition of common mental health disorders in Cambodia: Validation of the PHQ-9 and GAD-7 and development of a brief mental health screener.

PMEN-D-24-00347R1

Dear Dr. Steinman,

We are pleased to inform you that your manuscript 'Improving recognition of common mental health disorders in Cambodia: Validation of the PHQ-9 and GAD-7 and development of a brief mental health screener.' has been provisionally accepted for publication in PLOS Mental Health.

Best regards,

Haroon Lone

Academic Editor

PLOS Mental Health

Reviewer Comments (if any, and for reference):

Reviewer's Responses to Questions

**Comments to the Author**

1. If the authors have adequately addressed your comments raised in a previous round of review and you feel that this manuscript is now acceptable for publication, you may indicate that here to bypass the “Comments to the Author” section, enter your conflict of interest statement in the “Confidential to Editor” section, and submit your "Accept" recommendation.

Reviewer #1: All comments have been addressed

Reviewer #2: All comments have been addressed

2. Does this manuscript meet PLOS Mental Health’s publication criteria ? Is the manuscript technically sound, and do the data support the conclusions? The manuscript must describe methodologically and ethically rigorous research with conclusions that are appropriately drawn based on the data presented.

Reviewer #1: Yes

Reviewer #2: Yes

3. Has the statistical analysis been performed appropriately and rigorously?

Reviewer #1: Yes

Reviewer #2: Yes

4. Have the authors made all data underlying the findings in their manuscript fully available (please refer to the Data Availability Statement at the start of the manuscript PDF file)?

Reviewer #1: Yes

Reviewer #2: Yes

5. Is the manuscript presented in an intelligible fashion and written in standard English?

Reviewer #1: Yes

Reviewer #2: Yes

6. Review Comments to the Author

Reviewer #1: Thank you for addressing the comments! It flows very nicely.

Reviewer #2: Many thanks to authors for addressing all the concerns I remained previously.

7. PLOS authors have the option to publish the peer review history of their article (what does this mean? ). If published, this will include your full peer review and any attached files.

**Do you want your identity to be public for this peer review?** For information about this choice, including consent withdrawal, please see our Privacy Policy .

Reviewer #1: No

Reviewer #2: **Yes: ** Zhiyi Chen
